# METRO: MEMORY-ENHANCED TRANSFORMER FOR RETROSYNTHETIC PLANNING VIA REACTION TREE

## ABSTRACT

Retrosynthetic planning plays a critical role in drug discovery and organic chemistry. Starting from a target molecule as the root node, it aims to find a complete reaction tree subject to the constraint that all leaf nodes belong to a set of starting materials. The multi-step reactions are crucial because they determine the flow chart in the production of the Organic Chemical Industry. However, existing datasets lack curation of tree-structured multi-step reactions, and fail to provide such reaction trees, limiting models' understanding of organic molecule transformations. In this work, we first develop a benchmark curated for the retrosynthetic planning task, which consists of 124,869 reaction trees retrieved from the public USPTO-full dataset. On top of that, we propose Metro: **M**emory-**E**nhanced **T**ransformer for **R**etr**O**synthetic planning. Specifically, the dependency among molecules in the reaction tree is captured as context information for multi-step retrosynthesis predictions through transformers with a memory module. Extensive experiments show that Metro dramatically outperforms existing single-step retrosynthesis models by at least 10.7% in top-1 accuracy. The experiments demonstrate the superiority of exploiting context information in the retrosynthetic planning task. Moreover, the proposed model can be directly used for synthetic accessibility analysis, as it is trained on reaction trees with the shortest depths. Our work is the first step towards a brand new formulation for retrosynthetic planning in the aspects of data construction, model design, and evaluation.

## 1 INTRODUCTION

Retrosynthetic planning is a fundamental problem in organic chemistry (Coley et al., 2018a; Genheden et al., 2020). The goal of retrosynthetic planning is to find a series of starting molecules that go through a sequence of reactions, which can also be represented as reaction tree, to synthesize the target molecule. Retrosynthetic planning can be decomposed into multi-step retrosynthesis reactions through which we find all starting molecules that meet the requirements. The multi-step reactions outline the transformation direction of organic molecules and the transformation target. In the production of the Organic Chemical Industry, it requires us to design efficient organic synthesis routes to synthesize our desired target products at a low cost. Therefore, given a target molecule, predicting reasonable and efficient reaction routes to synthesize this molecule is a very crucial problem in both machine learning and organic chemistry (Segler et al., 2018).

To tackle this problem, past works, including MCTS (Segler et al., 2018), DFPN-E (Kishimoto et al., 2019), Retro*(Chen et al., 2020), self-improved retrosynthetic planning (Kim et al., 2021), RetroGraph (Xie et al., 2022), and Grasp (Yu et al., 2022), model the retrosynthetic planning as a search problem (Xie et al., 2022). Specifically, they first utilize reactions to train a template-based MLP retrosynthesis model (Segler et al., 2017) and then learn a search algorithm to perform a backward search to transform the molecules through retrosynthesis predictions until all the reactants are starting materials (Chen et al., 2020). The current benchmark for test evaluation of retrosynthetic planning models consists of 189 test routes (Chen et al., 2020).

These approaches have the following limitations: 1) the training dataset of single-step reactions limits the understanding of the transformation of organic molecules as a sequence of chaining chemical reactions. 2) past works use single-step retrosynthesis models, which neglect the context information in the reaction tree. 3) the test set is too small to comprehensively evaluate the performance. 4) the

evaluation unit of existing benchmark is the reaction route which is one path from the root node to the leaf node in the reaction tree.

In this work, we address these limitations by first constructing a new benchmark with 124,869 reaction trees retrieved from the public USPTO dataset and leverage the retrosynthesis transformer with an additional memory module to capture reaction tree information for retrosynthetic planning.

**Benchmark.** SCScore (Coley et al., 2018b) concludes that the number of steps required to synthesize a molecule is an accurate metric for estimating molecule synthetic accessibility. Based on this observation and inspired by the prediction of synthesis accessibility with reaction knowledge graph (Li & Chen, 2022), we construct a reaction graph from the existing reactions in the database. On the reaction graph, directed edges represent retrosynthesis reactions where the starting point denotes the product molecule and the ending point represents the reactant molecule to synthesize this product. Given a target molecule, we can search the shortest routes to form an efficient reaction tree from the reaction graph, while the ending points of these routes are the starting molecules that satisfy the requirements. By constructing the reaction trees for target molecules, we can obtain a new benchmark for our retrosynthetic planning task.

**Metro.** In this work, we propose **Metro**: **M**emory-**E**nhanced **T**ransformer for **R**etr**O**synthetic planning by extending Transformer with an additional memory module. Our proposed Metro can capture the dependency among the molecules on the reaction route as context information. By taking the context information into consideration, we can control the search within a reasonable reaction space specified for the reaction route. Extensive experimental results on retrosynthetic planning show that Metro achieves up to 13.2%, 14.5%, 11.1%, 10.5%, and 10.0% over transformer on top-1, top-2, top-3, top-4, and top-5 accuracy, which demonstrates the superiority of exploiting context information for retrosynthetic planning task.

## 2 Preliminaries

In this section, we formally define important terminologies used in the rest of the paper, including SMILES representation, starting material, and reaction tree.

**SMILES Representation.** The simplified molecule-input line-entry system (SMILES) (Weininger, 1988) is a chemical specification for describing the structure of chemical compounds using strings. Organic compounds can be denoted by SMILE representations like in Fig. 1, which is well suited for machine learning models to process. We denote the SMILES representation of molecule $x$ as $s(x)$, where $s(x)_i$ is the character at the $i$-th position of the string $s(x)$. Given a reaction $r_1 + r_2 + \ldots + r_n \to p$, the SMILES representation of this reaction is as follows:

$$s(r_1).s(r_2)\ldots s(r_n) \to s(p), \tag{1}$$

where multiple reactants are concatenated by "." in the SMILES representation.

**Starting Material.** We denote the space of all chemical molecules as $\mathcal{M}$. The starting materials are a special set of molecules, denoted as $\mathcal{S} \subseteq \mathcal{M}$. AiZynthFinder (Genheden et al., 2020) defines the starting material as a commercially purchasable compound. ZINC (Sterling & Irwin, 2015) releases the open source databases of purchasable compounds. We define this list of compounds in these databases as our starting materials.

**Reaction Tree and Reaction Routes.** Given the above definitions, we can denote a reaction tree (Shibukawa et al., 2020; Nguyen & Tsuda, 2021) as $\mathcal{T} = \{T, \mathcal{R}, \mathcal{I}, \tau\}$, where $T \in \mathcal{M} \setminus \mathcal{S}$ is the product molecule we desire to synthesize (A in Fig. 1), $\mathcal{R} = \{r_1, r_2, \ldots, r_n\} \subseteq \mathcal{S}$ is the set of starting materials (E, F, G, H in Fig. 1) that go through a series of reactions $\tau$ to synthesize A, and $\mathcal{I} = \{m_1, m_2, \ldots, m_u\} \subseteq \mathcal{M} \setminus \mathcal{S}$ is the set of intermediate products (B, C, D in Fig. 1) where intermediate products are formed from reactants or intermediate products and then react further to give the final product or produce intermediate products. A reaction tree consists of multiple reaction routes. A reaction route is a path from the target molecule to a starting material in the reaction tree. According to the definition, the number of reaction routes is equal to the number of starting materials. We denote reaction route as $l$, the set of reaction routes as $\mathcal{L} = \{l_1, l_2, \ldots, l_n\}$, and we have

$$\tau = \tau_{l_1} \cup \tau_{l_2} \cup \cdots \cup \tau_{l_n}, \tag{2}$$

O=C(Cl)c1cc(Cl)cc(Cl)c1Cl

C

O=C(Cl)c1cc(Cl)cc(Cl)c1Cl>
>O=C(O)c1cc(Cl)cc(Cl)c1Cl

G

c1ccc2c(c1)CNc1ccccc1C2

F

[O-][S+](Cl)Cl

H

A

O=C(Nc1ccc(C(=O)N2Cc3ccccc3Cc
3ccccc32)cc1)c1cc(Cl)cc(Cl)c1Cl

Nc1ccc(C(=O)N2Cc3cc
ccc3Cc3ccccc32)cc1

B

O=C(c1ccc([N+](=O)[O-
])cc1)N1Cc2ccccc2Cc2ccccc21

D

O=C(Cl)c1ccc([N+](=O)
[O-])cc1

E

Figure 1: Reaction tree. Given the definition in Eq. (3), the depth of this tree is 3, which means the depth of the longest reaction route is 3. A is the desired product molecule to be synthesized. B, C, and D are the intermediate product molecules. E, F, G, and H are the starting molecules.

where $\tau_{l_i}$ is the set of reactions accompanying this reaction route $l_i$. As illustrated in Fig. 1, A->B->D->E is one of the reaction routes in this tree. We denote the depth $\mathcal{D}_{\mathcal{T}}$ of a reaction tree as the length of the longest reaction route in this tree, where

$$\mathcal{D}_{\mathcal{T}} = \max_i \mathcal{D}_{l_i}. \tag{3}$$

The depth of a reaction tree is also the number of steps required to synthesize a molecule from a fixed set of commercially purchasable compounds. Note that in this paper, the default order of the reaction route is the retrosynthesis order.

## 3 PROBLEM FORMULATION OF RETROSYNTHETIC PLANNING

In this section, we formally give the problem formulation and the goal of retrosynthetic planning.

**Single-Step Retrosynthesis.** Given a target product molecule $T \in \mathcal{M}$, the goal of retrosynthesis is to predict a set of reactants $\mathcal{R} = \{r_1, r_2, \ldots, r_n\} \subseteq \mathcal{M}$ that can react to synthesize this product, which can be formulated as:

$$T \to \mathcal{R}.$$

**Retrosynthetic Planning.** Given a target molecule $T \in \mathcal{M}$, the goal of retrosynthetic planning is to search for the starting materials $\mathcal{R} = \{r_1, r_2, \ldots, r_n\} \subseteq \mathcal{S}$ that can synthesize the target molecule through a set of chemical reactions $\tau = \{R_1, R_2, \ldots, R_m\}$, which can be formulated as follows:

$$T \to \mathcal{I} \to \mathcal{R}, \tag{4}$$

where $\mathcal{I} \subseteq \mathcal{M} \setminus \mathcal{S}$ is the set of intermediate product molecules.

**The Goal of Retrosynthetic Planning.** Our goal of retrosynthetic planning is to find the reaction tree with minimum depth to synthesize the target molecule. The reaction routes for retrosynthetic planning we construct follow the principle of finding starting materials as early as possible. This principle guides the transformation direction of the molecules, thus enabling our machine learning models to make predictions on the best retrosynthesis direction. Moreover, what needs to be declared is that molecule synthesis accessibility is a part of our work due to the construction of the dataset. By predicting the reaction tree with minimum depth, we can also make a prediction of the number of steps needed to synthesize a target molecule.

## 4 NEW BENCHMARK BASED ON REACTION TREES

In this section, we describe the details of how to construct the new benchmark. Current benchmark (Chen et al., 2020) has the two limitations: 1) the test size is too small to evaluate the performance of models, 2) the dataset is based on reaction route instead of reaction tree. Therefore, we build a new benchmark based on reaction trees. The benchmark construction consists of three steps: reaction graph construction, reaction tree traversal, and dataset split.

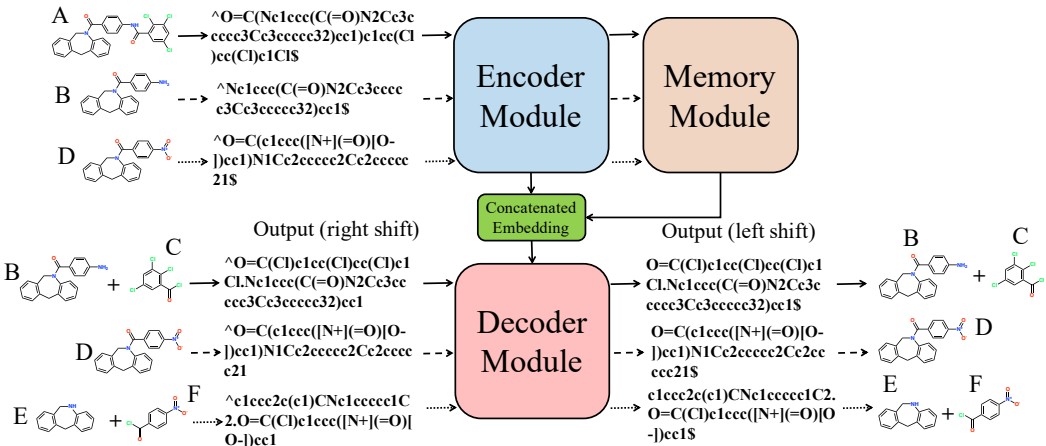

Figure 2: Model overview. Our model consists of three modules: encoder, memory module, and decoder. The three styles of → mean the three different retrosynthesis reactions on the reaction route.

**Reaction Graph Construction.** We extract all the reactions from USPTO-all dataset (Lowe, 2012) and build a directed reaction graph. This is partially inspired by the construction of the reaction knowledge graph in (Li & Chen, 2022), which is used to identify the minimum steps needed to synthesize molecules for synthesis accessibility analysis. Let $\mathcal{G} = \{\mathcal{V}, \mathcal{E}\}$ denote the directed graph, where $\mathcal{V}$ is set of nodes $\{v_1, v_2, \cdots, v_N\}$ with $|\mathcal{V}| = N$ and $\mathcal{E}$ is the set of edges. On this graph $\mathcal{G}$, $v_i$ is the molecule that exists in the reaction database and $(v_i, v_j)(v_i \rightarrow v_j)$ is the directed edge which means $v_i$ and $v_j$ exists in the same retrosynthesis reaction where $v_i$ is the product molecule and $v_j$ is one of the reactants. One molecule can be a product or reactant on the graph.

**Reaction Tree Traversal.** We perform reaction tree traversal after reaction graph construction. We treat the nodes whose in-degree is 0 as our desired product molecules, which can be represented as $\mathcal{V}_p$. Given one node $v \in \mathcal{V}_p$, we perform dynamic programming and backtracking to find all the reaction trees $\{\mathcal{T}_v^{(1)}, \mathcal{T}_v^{(2)}, \cdots, \mathcal{T}_v^{(s)}\}$ where the leaf nodes are starting materials for $v$. For two trees which have different retrosynthesis reactions for the same product, we think these are different trees.

**Dataset Split.** We first build the reaction graph based on all the reactions in the database, in order to identify reaction trees with smaller depth (Chen et al., 2020). We then split the reaction trees into training/validation/test datasets. This procedure aligns with the scenario in real-world organic chemical production, where we have access to a huge database of reactions and we can find the reaction tree with the shortest depth with our dataset construction method. Although the reaction tree of the training and test datasets will have overlapping reactions, we only select the target molecule that does not exist in the training dataset in the testing phase and do not provide any information about the reaction tree of this target molecule to avoid information leakage. Moreover, many organic synthesis routes share basic chemical reactions. Therefore, it is reasonable that the reaction trees in the training and test datasets have overlapping reactions. Note that there may be several reactions to synthesize a target molecule, leading to multiple reaction trees. We extract the reaction trees with the minimum depth as our dataset according to the synthetic accessibility criteria (Li & Chen, 2022).

## 5 METRO

In this section, we describe the details of our proposed Metro. Our model is based on single-step retrosynthesis transformer (Vaswani et al., 2017; Karpov et al., 2019). We further propose a memory module that can capture the dependencies between molecules on the retrosynthesis route as context information. We don't rely on a template to formulate the reaction pattern or labeling number (Lin et al., 2022) to extract the reaction center. The training and inference can be done in an end-to-end manner. Fig. 2 outlines the sketch of our model.

### 5.1 Overview of Metro

#### 5.1.1 Architecture

As shown in Fig .2, our model consists of three components: encoder, memory module, and decoder. The input for the encoder is the SMILES representations of the product molecules (A, B, D) on the same reaction route. The SMILES representations of molecules have been processed. We add the start symbol $\wedge$ and the end symbol $\$$ for the input. We also put the output in the decoder. But there is a difference between the output SMILES fed into the decoder and the output SMILES predicted by the decoder. We add the start symbol $\wedge$ (right shift) on the former output SMILES and the end symbol $\$$ (left shift) on the later output SMILES. So there is not any leakage problem with the supervised information. The encoder transforms the embedding matrices of product molecules into latent representations (matrices), which is also called the *encoder output*. We take the encoder output as input to obtain the context information (memory output) through the memory module. After that, we concatenate the encoder output and memory output and feed it into the decoder.

#### 5.1.2 Sequence Generation

Metro models the prediction of reaction route as a sequence of molecule generation. Given the SMILES representations of (A, B, D) from the same reaction route as input, the output of our model is the SMILES representations of (B+C, D, E+F). When we predict B+C, the input is A and we will not consider the information of (B, D). When we complete the prediction of (B, C), we add B to the reaction route and use (A, B) as input to predict D. At the same time, we also add C to another reaction route as input to predict (G, H) as shown in Fig. 1. We repeat this step until all reactants are starting materials. So during the training stage, we do not consider the information of later molecules when making the current prediction. The attention mechanism ensures that our predictions of all reactions along the same route are parallelized at the training stage. This can be implemented by masking the input of later molecules.

### 5.2 Memory Module

In this section, we will describe the details of the memory module. We employ attention mechanism (Vaswani et al., 2017) and memory network (Weston et al., 2015; Sukhbaatar et al., 2015; Ramsauer et al., 2020) to build our memory layer to capture the context information on the reaction route. We formally describe the blocking components of one memory layer as follows. Our memory module takes a series of encoder output $X_1, X_2, \ldots, X_n$ as input, where $n$ is the depth of the reaction route. Each of $X_i \in \mathbb{R}^{l \times d}$ is the latent embedding matrix corresponding to the $i$-th molecule of the encoder input (A, B, and D in our illustrated case). We first use the Linear Projection of Flattened Patches employed in ViT (Dosovitskiy et al., 2020) to transform matrices $X_1, X_2, \ldots, X_n$ into vectors $v_1, v_2, \ldots, v_n$ as follows:

$$v_i = X_i W_p, \tag{5}$$

where $W_p \in \mathbb{R}^{(l \cdot d) \times d}$, and $l$ is the length of SMILES. We then use $v_i$ to compute the query and key and use $X_i$ to compute the value as follows:

$$\begin{aligned}
Q &= [v_1, v_2, \cdots, v_n]^T W^Q \\
K &= [v_1, v_2, \cdots, v_n]^T W^K \\
V &= [X_1, X_2, \cdots, X_n]^T W^V
\end{aligned} \tag{6}$$

where $W^Q \in \mathbb{R}^{d \times d_k}$ is the query matrix, $W^K \in \mathbb{R}^{d \times d_k}$ is the key matrix, and $W^V \in \mathbb{R}^{d \times d_k}$ is the value matrix. Then we compute the memory embedding matrices as follows:

$$\text{Attention}(Q, K, V) = \text{softmax}\left(\frac{QK^T}{\sqrt{d_k}}\right) V \tag{7}$$

The output $\text{Attention}(Q, K, V)$ is computed as a weighted sum of the values $V$, where the weight assigned to each value is the attention score between two molecules.

**Retrosynthetic-Planning Architecture Design.** The difference between our attention mechanism in our memory layer with that of standard transformers is that we compute the attention scores of

the two embedding matrices instead of vectors due to the embedding matrix representation of the molecules. We also utilize multi-head attention to jointly compute the context information from different subspaces and then we use residual connection (He et al., 2016) and fully connected layer (FFN) to get the memory output $M_1, M_2, \ldots, M_n$. We treat the memory output $[M_1, M_2, \cdots, M_n]^T$ as context information. $M_i$ captures the dependencies between $X_i$ and $X_1, X_2, \ldots, X_{i-1}$. After getting the embedding matrices of context information, we concatenate them with the encoder output to feed the decoder.

## 5.3 WHY CONTEXT INFORMATION

The intuition of exploiting the reaction tree (context information) instead of single-step prediction is that we can better prune the reaction search space. The single-step retrosynthesis reaction model searches the entire chemical reaction space of a molecule, but some candidates, although valid, do not align with the synthesis goal of the current synthetic route and should be discarded when taking the entire reaction tree into account. Experimental results show that our proposed Metro improves by a large margin over Transformer.

## 5.4 INFERENCE

In the inference stage, we start from the target molecule $T$, and perform backward chaining to do a series of one-step retrosynthesis predictions until the reactants are all starting materials. This backward method is also adopted by Chen et al. (2020). After getting the predicted reactant molecules for the retrosynthesis reaction at each step, we refer to the set of starting materials to check whether the reactant molecules are starting materials. If they are starting materials, we add them into the prediction reactant set. Otherwise, we get a new reaction route and predict the next step's output. Once we obtain the predicted reactant set, we compare it with the ground truth reactant set and get the inference accuracy. The inference process is outlined in the Algorithm 1. Note that we perform a Depth First Search (DFS) for all models in our paper.

---

**Algorithm 1** Inference of the reaction tree given a target molecule

---

1: **Input:** Target molecule $T$, starting material set $\mathcal{S}$
2: Initialize reactant set $\mathcal{R} = \{\}$, reaction route set $\mathcal{L} = \{\}$
3: Put the initial route $[T]$ into $\mathcal{L}$
4: **while** $\mathcal{L}$ is not a empty set **do**
5:     Take an route $l$ from $\mathcal{L}$
6:     Predict the reactants $r_l$ given $l$
7:     **for** reactant $r_l^{(i)}$ in $r_l$ **do**
8:         **if** $r_l^{(i)} \in \mathcal{S}$ **then**
9:             Put $r_l^{(i)}$ into $\mathcal{R}$
10:         **else**
11:             Generate a new rote $l' = l + [r_l^{(i)}]$
12:             Put $l'$ into $\mathcal{L}$
    **return** predicted reactant set $\mathcal{R}$

---

# 6 EXPERIMENTS

In this section, we evaluate the performance of different base models on our proposed benchmark for the retrosynthetic planning task.

## 6.1 EXPERIMENT SETUP

**Dataset.** We utilize the public dataset USPTO-full to construct the benchmark for the retrosynthetic planning task. The USPTO-full dataset consists of 1,808,937 reactions. After removing invalid and duplicate reactions, we obtain 906,164 reactions. Based on these reactions, we construct a reaction graph. We treat the molecules whose out-degree is 0 as our desired target molecules. And we use dynamic programming and backtracking to find all the reaction trees for each target molecule. There are 124,869 molecules for which we can find the reaction trees where the leaf nodes are all starting materials. We extract molecules with shortest depths greater than 1 and split these molecules into training, validation, and test datasets. For those molecules with shortest depths between 2 and 10, they are randomly split into training/validation/test datasets following 80%/10%/10% proportions. For those molecules with shortest depths larger than 10, we put them into the test dataset to evaluate the performance of models on the molecules which need too many steps to synthesize. The number

Table 1: Top-$k$ exact match accuracy.

| Methods | Top-$k$ accuracy % | | | | |
|---|---|---|---|---|---|
| | 1 | 2 | 3 | 4 | 5 |
| *Template-based* | | | | | |
| RETROSIM (Coley et al., 2017) | 23.2 | 27.2 | 28.9 | 30.0 | 30.5 |
| NEURALSYM (Segler & Waller, 2017) | 26.8 | 32.2 | 34.1 | 35.2 | 35.8 |
| GLN (Dai et al., 2019) | 25.9 | 32.7 | 35.0 | 36.5 | 37.2 |
| *Semi-template-based* | | | | | |
| G2Gs (Shi et al., 2020) | 4.0 | 6.1 | 7.2 | 8.2 | 8.8 |
| GRAPHRETRO (Somnath et al., 2021) | 14.4 | - | - | - | - |
| *Template-free* | | | | | |
| TRANSFORMER (Karpov et al., 2019) | 24.3 | 33.1 | 37.2 | 39.7 | 41.2 |
| MEGAN (Sacha et al., 2021) | 20.1 | 29.5 | 34.9 | 38.3 | 40.4 |
| **METRO (Ours)** | **37.5** | **47.6** | **48.3** | **50.2** | **51.2** |

of samples in training/validation/test datasets are 46,458, 5,803, and 5,838 respectively. The dataset statistics can be found in Appendix A.

**Evaluation Protocol.** Inspired by the evaluation metric of single-step retrosynthesis reaction, we decide whether a prediction is accurate by comparing our predicted starting material set with the ground truth starting material set is an exact match. Note that for a specific target molecule, there may be multiple reaction trees in the test set. It is an accurate match when the predicted starting material set can hit one of the multiple ground truths. Besides, we perform a pruning search, when the length of the reaction route predicted by the model exceeds the depth of the ground truth reaction tree, we stop the search. By using our evaluation metric, we can obtain the test accuracy of the our model and the baselines, laying a benchmark for future works and enabling intuitive comparison.

**Baselines.** We focus on the evaluation of base models on the retrosynthetic planning task, and adopt the DFS algorithm to perform the search to demonstrate the immediate gain of our model. One future work is to combine and compare with other retrosynthetic planning search algorithms. We evaluate our proposed method against existing single-step retrosynthesis models, which can be classified into three categories: template-based, template-free, and semi-template-Based. All single-step methods are trained on the single-step retrosynthesis reactions independently in the training dataset. When training is completed, we perform DFS to search the reactants as in Algorithm 1. Note that the single-step models only rely on the last product molecule on the reaction route $l$ to make the prediction. For the baselines, we follow their experimental setup such as hyper-parameter and data processing in their paper and conduct the experiments with their released codes except for TRANSFORMER[1].

*Template-Based*: RETROSIM (Coley et al., 2017) computes similarity between the fingerprints of product and reactants in the training dataset to rank the templates for a given target molecule. NEURALSYM (Segler & Waller, 2017) trains a MLP to model the template selection as a classification problem. GLN (Dai et al., 2019) exploits a conditional graphical model to build the connection between probabilistic models and reaction templates.

*Template-Free*: TRANSFORMER (Karpov et al., 2019) models the retrosynthesis prediction as a sequence-to-sequence problem based on Transformer (Vaswani et al., 2017). MEGAN (Sacha et al., 2021) models the retrosynthesis as a sequence of graph edits.

*Semi-Template-Based*: G2Gs (Shi et al., 2020) and GRAPHRETRO (Somnath et al., 2021) are two-stage models, which first identify the reaction center, break the products into synthons, and then expand the synthons into reactants. G2Gs model the expansion of synthons to reactants as a sequential generation of atoms and bonds. GRAPHRETRO exploits the leaving groups extracted from dataset to model the expansion as a classification problem.

**Implementation Details.** Based on the vanilla TRANSFORMER (Karpov et al., 2019), we only introduce additional 3 memory layers and do not touch other components. For the hyper-parameters,

---

[1]We implement TRANSFORMER using Pytorch (Paszke et al., 2019).

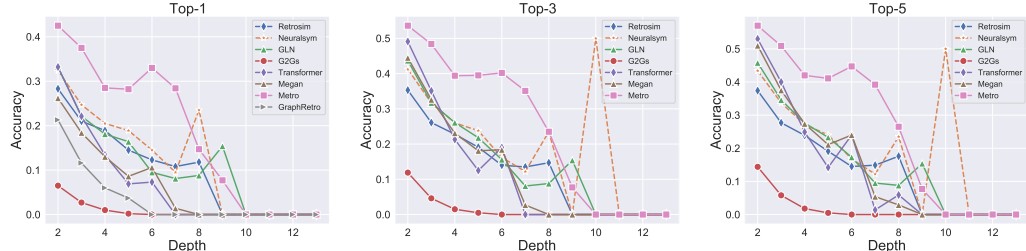

Figure 3: The top-1, top-3, and top-5 test accuracy in terms of depth (the minimum steps required to synthesize a target molecule). There is no beam search for GRAPHRETRO because of missing probabilities of each answer in their released codes. The cases whose depth is 10 have too few data points for the results to be convincing, although there is a jump at depth 10 for NEURALSYM.

we directly followed the reported setting in their released codes, and did not perform additional hyperparameter tuning. We still achieve a dramatic improvement for the planning task, which well demonstrates the effectiveness of our proposed method. The details of hyper-parameters can also be found in Appendix B.1. METRO is trained on 2 NVIDIA Tesla V100 GPUs.

## 6.2 RESULTS

**Main Results.** Table 1 reports the main results. From the table, we observe that our proposed METRO achieves the best performance and outperforms the baselines by at least 10.7% of top-1 test accuracy. Besides, the results demonstrate that our proposed model achieves a better performance against TRANSFORMER. More specifically, we can improve upon TRANSFORMER by a margin of 13.2%, 14.5%, 11.1%, 10.5%, and 10.0% on top-1, top-2, top-3, top-4, and top-5 accuracy respectively. It demonstrates that the context information on the reaction route captured by our introduced memory module enables our model to search the reactants for single-step retrosynthesis reaction in a reasonable reaction space specified for this reaction route. Context information also preventing us searching through the whole reaction space, thus avoiding some wrong answers. Moreover, the results of baselines on our retrosynthetic planning task do not match well with single-step retrosynthesis prediction. Existing semi-template-based models outperform or match the template-based and template-free models on the single-step retrosynthesis prediction but have poor performance on our task. A reasonable reason is that approximately 95% of the reactions in the USPTO-50K dataset used for single-step retrosynthesis reactions have only one reaction center. But in our constructed dataset, approximately 30% of reactions have multiple reaction centers. G2GS only can handle cases with one reaction center, and so perform badly on our task. Template-free models, which do not need to extract templates or find the reaction center using the labeling number, model the retrosynthesis as a sequence-to-sequence problem or a sequence of graph edits, which has better scalability and perform better on retrosynthetic planning tasks.

**Synthetic Accessibility Analysis.** We group by the length of the reaction tree of the target molecules and report the top1, top3, and top5 test accuracy for each group. We report the results in Fig 3. The results show that the more steps required to synthesize a target molecule, the lower the accuracy of all models' predictions. When the number of steps required is greater than 9, our model cannot predict the starting materials to synthesize the target molecule. This result is also consistent with molecule synthetic accessibility. The molecules are hard to synthesize when the number of reaction steps required is larger in Li & Chen (2022). Our results further support this view. From Fig 3, we can observe that our model still performs better than all baselines when predicting the starting materials of the target molecules which need 2-7 reaction steps to synthesize. When testing on the 49 cases which need 8-10 steps, only GLN and NEURALSYM outperform or match our model. When testing on cases that need more than 10 steps, all models can not make an accurate prediction. One future work is to add the template information to our model to improve the performance of the cases which need too many steps to synthesize.

## 6.3 CASE STUDY

In Fig. 4, we visualize the predictions from METRO and TRANSFORMER. The top of Fig. 4 are the correct predictions from METRO, and the bottom are the wrong predictions from TRANSFORMER.

Figure 4: **Case Study.** We split the predicted reaction tree into retrosynthesis reactions. On the top is the correct reaction tree predicted by METRO, and on the bottom is the predicted reaction tree predicted by TRANSFORMER.

From Figure 4, we can observe that TRANSFORMER makes a wrong prediction on the third-step retrosynthesis reaction. Since the whole reaction space on which TRANSFORMER searches is too large to get correct predictions. Due to the dependency of molecules on the reaction route as context information, our METRO can search the reactants on a reasonable space. So our model can make correct predictions.

## 7 RELATED WORK

**Retrosynthesis Model.** Existing machine learning models for retrosynthesis prediction can be classified into Template-based, Semi-template-based, and Template-free models. Template-based retrosynthetic algorithms (Chen et al., 2020; Coley et al., 2017; Dai et al., 2019; Segler & Waller, 2017; Chen & Jung, 2021; Seidl et al., 2021) extract patterns from the training data which encode how atoms and bonds change during the reaction. Semi-template-based models (Shi et al., 2020; Yan et al., 2020; Somnath et al., 2021) predict reactants via two stages: reaction center identification and reactants generation. Template-free algorithms (Liu et al., 2017; Zheng et al., 2019; Chen et al., 2019; Karpov et al., 2019) model the retrosynthesis as a sequence-to-sequence problem. Our work is closely related to the single-step retrosynthesis transformer. To capture the context information of the reaction route, we introduce the memory module (Sukhbaatar et al., 2015).

**Retrosynthetic Planning Search Algorithm.** Existing Retrosynthetic Planning Search Algorithms model the retrosynthetic planning as a search problem. MCTS (Segler et al., 2018) employs Monte Carlo tree search, DFPN-E (Kishimoto et al., 2019) combines Depth-First Proof-Number (DFPN) with Heuristic Edge Initialization, Retro* (Chen et al., 2020) proposes a neural-based A*-like algorithm, and RetroGraph (Xie et al., 2022) proposes a graph-based search policy. Our solution focuses on the base model which differs greatly from search algorithms.

**Synthetic Accessibility** Some machine learning models have been proposed for the estimation of the synthetic difficulty of molecules, such as SAscore (Ertl & Schuffenhauer, 2009), PGFS (Gottipati et al., 2020), SCScore (Coley et al., 2018b), SYBA (Voršilák et al., 2020), RAscore (Thakkar et al., 2021), and knowledge graph (Li & Chen, 2022). These approaches divide drug molecules into easy and hard-to-synthesize categories, where hard-to-synthesize molecules require longer reaction routes to synthesize.

## 8 CONCLUSION

In this work, we build a reaction-tree-based benchmark. We also propose a new retrosynthetic planning base model by extending the single-step retrosynthesis transformer with an additional memory module. Our memory module can capture the context information of the reaction route, enabling us to make a better prediction for the retrosynthesis reaction on this route.

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

## A    DATASETS DETAILS

Table 2: The number of target molecules in training/validation/test datasets in term of the shortest depths to synthesize the target molecules.

| #Molecules \ Depth Dataset | 2 | 3 | 4 | 5 | 6 | 7 | 8 | 9 | 10 | 11 | 12 | 13 |
|---|---|---|---|---|---|---|---|---|---|---|---|---|
| Training | 22,903 | 12,004 | 5,849 | 3,268 | 1,432 | 594 | 276 | 107 | 25 | 0 | 0 | 0 |
| Validation | 2,862 | 1,500 | 731 | 408 | 179 | 74 | 34 | 13 | 2 | 0 | 0 | 0 |
| Test | 2,862 | 1,500 | 731 | 408 | 179 | 74 | 34 | 13 | 2 | 32 | 2 | 1 |

## B    REPRODUCIBILITY

### B.1    IMPLEMENTATION DETAILS

We use Pytorch (Paszke et al., 2019) to implement Metro. The codes of baselines are implemented referring to the implementation of RETROSIM[2], NEURALSYM[3], GLN[4], G2Gs[5], GRAPHRETRO[6], TRANSFORMER[7], and MEGAN[8]. All the experiments in this work are conducted on a single NVIDIA Tesla V100 with 32GB memory size. The software that we use for experiments are Python 3.6.8, pytorch 1.9.0, pytorch-scatter 2.0.9, pytorch-sparse 0.6.12, numpy 1.19.2, torchvision 0.10.0, CUDA 10.2.89, CUDNN 7.6.5, einops 0.4.1, and torchdrug 0.1.3.

### B.2    HYPERPARAMETER DETAILS

Table 3: The hyper-parameters for Metro.

| | |
|---|---|
| max length | 200 |
| embedding size | 64 |
| encoder layers | 3 |
| decoder layers | 3 |
| memory layers | 3 |
| attention heads | 10 |
| FFN hidden | 512 |
| dropout | 0.1 |
| epochs | 4000 |
| batch size | 64 |
| warmup | 16000 |
| lr factor | 20 |

## C    DETAILS OF SYNTHETIC ACCESSIBILITY ANALYSIS

---

[2] https://github.com/connorcoley/retrosim
[3] https://github.com/linminhtoo/neuralsym
[4] https://github.com/Hanjun-Dai/GLN
[5] https://torchdrug.ai/docs/tutorials/retrosynthesis
[6] https://github.com/vsomnath/graphretro
[7] https://github.com/bigchem/synthesis
[8] https://github.com/molecule-one/megan

Table 4: Top-$k$ exact match accuracy of RETROSIM in terms of depth of reaction tree.

| Depth | Top-$k$ accuracy (%) | | | | |
|---|---|---|---|---|---|
| | 1 | 2 | 3 | 4 | 5 |
| 2 | 28.3 | 33.2 | 35.3 | 36.7 | 37.4 |
| 3 | 21.0 | 24.5 | 26.1 | 27.1 | 27.7 |
| 4 | 18.9 | 22.0 | 22.8 | 23.9 | 23.9 |
| 5 | 14.5 | 17.2 | 19.1 | 19.1 | 19.1 |
| 6 | 12.3 | 14.0 | 14.0 | 14.5 | 14.5 |
| 7 | 10.8 | 13.5 | 13.5 | 14.9 | 14.9 |
| 8 | 11.8 | 11.8 | 14.7 | 17.6 | 17.6 |
| 9 | 0 | 0 | 0 | 0 | 0 |
| 10 | 0 | 0 | 0 | 0 | 0 |
| 11 | 0 | 0 | 0 | 0 | 0 |
| 12 | 0 | 0 | 0 | 0 | 0 |
| 13 | 0 | 0 | 0 | 0 | 0 |

Table 5: Top-$k$ exact match accuracy of NEURALSYM in terms of depth of reaction tree.

| Depth | Top-$k$ accuracy (%) | | | | |
|---|---|---|---|---|---|
| | 1 | 2 | 3 | 4 | 5 |
| 2 | 32.4 | 39.0 | 41.2 | 42.5 | 43.4 |
| 3 | 24.7 | 30.1 | 31.7 | 33.0 | 33.7 |
| 4 | 20.5 | 24.4 | 26.0 | 26.7 | 27.2 |
| 5 | 18.9 | 22.5 | 23.8 | 24.0 | 24.0 |
| 6 | 14.5 | 15.1 | 16.2 | 16.8 | 16.8 |
| 7 | 9.5 | 10.8 | 12.2 | 12.2 | 12.2 |
| 8 | 23.5 | 23.5 | 23.5 | 23.5 | 23.5 |
| 9 | 0 | 0 | 0 | 0 | 0 |
| 10 | 0 | 0 | 0 | 0 | 0 |
| 11 | 0 | 0 | 0 | 0 | 0 |
| 12 | 0 | 0 | 0 | 0 | 0 |
| 13 | 0 | 0 | 0 | 0 | 0 |

Table 6: Top-$k$ exact match accuracy of GLN in terms of depth of reaction tree.

| Depth | Top-$k$ accuracy (%) | | | | |
|---|---|---|---|---|---|
| | 1 | 2 | 3 | 4 | 5 |
| 2 | 33.2 | 41.1 | 43.7 | 45.1 | 45.8 |
| 3 | 22.1 | 29.7 | 31.7 | 33.7 | 34.6 |
| 4 | 18.1 | 23.1 | 26.1 | 26.9 | 27.6 |
| 5 | 16.4 | 20.8 | 21.8 | 23.0 | 23.3 |
| 6 | 9.5 | 12.8 | 15.6 | 16.8 | 17.3 |
| 7 | 8.1 | 8.1 | 8.1 | 8.1 | 9.5 |
| 8 | 8.8 | 8.8 | 8.8 | 8.8 | 8.8 |
| 9 | 15.4 | 15.4 | 15.4 | 15.4 | 15.4 |
| 10 | 0 | 0 | 0 | 0 | 0 |
| 11 | 0 | 0 | 0 | 0 | 0 |
| 12 | 0 | 0 | 0 | 0 | 0 |
| 13 | 0 | 0 | 0 | 0 | 0 |

Table 7: Top-$k$ exact match accuracy of G2GS in terms of depth of reaction tree.

| Depth | Top-$k$ accuracy (%) | | | | |
|---|---|---|---|---|---|
| | 1 | 2 | 3 | 4 | 5 |
| 2 | 6.5 | 10.1 | 11.9 | 13.5 | 14.4 |
| 3 | 2.7 | 3.9 | 4.6 | 5.1 | 5.8 |
| 4 | 1.0 | 1.1 | 1.5 | 1.8 | 1.8 |
| 5 | 0.2 | 0.5 | 0.5 | 0.5 | 0.5 |
| 6 | 0 | 0 | 0 | 0 | 0 |
| 7 | 0 | 0 | 0 | 0 | 0 |
| 8 | 0 | 0 | 0 | 0 | 0 |
| 9 | 0 | 0 | 0 | 0 | 0 |
| 10 | 0 | 0 | 0 | 0 | 0 |
| 11 | 0 | 0 | 0 | 0 | 0 |
| 12 | 0 | 0 | 0 | 0 | 0 |
| 13 | 0 | 0 | 0 | 0 | 0 |

Table 8: Top-$k$ exact match accuracy of GRAPHRETRO in terms of depth of reaction tree.

| Depth | Top-$k$ accuracy (%) | | | | |
|---|---|---|---|---|---|
| | 1 | 2 | 3 | 4 | 5 |
| 2 | 21.3 | - | - | - | - |
| 3 | 11.6 | - | - | - | - |
| 4 | 6.0 | - | - | - | - |
| 5 | 3.7 | - | - | - | - |
| 6 | 0 | - | - | - | - |
| 7 | 0 | - | - | - | - |
| 8 | 0 | - | - | - | - |
| 9 | 0 | - | - | - | - |
| 10 | 0 | - | - | - | - |
| 11 | 0 | - | - | - | - |
| 12 | 0 | - | - | - | - |
| 13 | 0 | - | - | - | - |

Table 9: Top-$k$ exact match accuracy of MEGAN in terms of depth of reaction tree.

| Depth | Top-$k$ accuracy (%) | | | | |
|---|---|---|---|---|---|
| | 1 | 2 | 3 | 4 | 5 |
| 2 | 26.2 | 37.8 | 44.5 | 48.6 | 51.0 |
| 3 | 18.4 | 27.3 | 32.6 | 35.3 | 37.5 |
| 4 | 13.0 | 20.0 | 23.1 | 26.1 | 27.4 |
| 5 | 8.6 | 14.5 | 18.1 | 19.9 | 21.1 |
| 6 | 10.6 | 14.5 | 18.4 | 21.8 | 24.0 |
| 7 | 1.4 | 1.4 | 2.7 | 5.4 | 5.4 |
| 8 | 0 | 0 | 0 | 0 | 2.9 |
| 9 | 0 | 0 | 0 | 0 | 0 |
| 10 | 0 | 0 | 0 | 0 | 0 |
| 11 | 0 | 0 | 0 | 0 | 0 |
| 12 | 0 | 0 | 0 | 0 | 0 |
| 13 | 0 | 0 | 0 | 0 | 0 |

Table 10: Top-$k$ exact match accuracy of TRANSFORMER in terms of depth of reaction tree.

| Depth | Top-$k$ accuracy (%) | | | | |
|---|---|---|---|---|---|
| | 1 | 2 | 3 | 4 | 5 |
| 2 | 33.2 | 44.4 | 49.1 | 51.6 | 53.0 |
| 3 | 22.1 | 30.8 | 35.1 | 38.4 | 40.0 |
| 4 | 13.4 | 17.9 | 21.3 | 23.3 | 24.9 |
| 5 | 6.9 | 10.3 | 12.5 | 13.5 | 14.2 |
| 6 | 7.3 | 14.5 | 19.0 | 22.3 | 24.0 |
| 7 | 0 | 0 | 0 | 1.4 | 1.4 |
| 8 | 0 | 0 | 0 | 2.9 | 5.9 |
| 9 | 0 | 0 | 0 | 0 | 0 |
| 10 | 0 | 0 | 0 | 0 | 0 |
| 11 | 0 | 0 | 0 | 0 | 0 |
| 12 | 0 | 0 | 0 | 0 | 0 |
| 13 | 0 | 0 | 0 | 0 | 0 |

Table 11: Top-$k$ exact match accuracy of METRO in terms of depth of reaction tree.

| Depth | Top-$k$ accuracy (%) | | | | |
|---|---|---|---|---|---|
| | 1 | 2 | 3 | 4 | 5 |
| 2 | 42.5 | 50.2 | 53.6 | 55.9 | 57.0 |
| 3 | 37.5 | 45.2 | 48.4 | 50.0 | 50.9 |
| 4 | 28.5 | 36.5 | 39.4 | 40.8 | 42.0 |
| 5 | 28.2 | 35.5 | 39.5 | 40.4 | 41.1 |
| 6 | 33.0 | 39.1 | 40.2 | 43.6 | 44.7 |
| 7 | 28.4 | 32.4 | 35.1 | 39.2 | 39.2 |
| 8 | 14.7 | 17.6 | 23.5 | 23.5 | 26.5 |
| 9 | 7.7 | 7.7 | 7.7 | 7.7 | 7.7 |
| 10 | 0 | 0 | 0 | 0 | 0 |
| 11 | 0 | 0 | 0 | 0 | 0 |
| 12 | 0 | 0 | 0 | 0 | 0 |
| 13 | 0 | 0 | 0 | 0 | 0 |

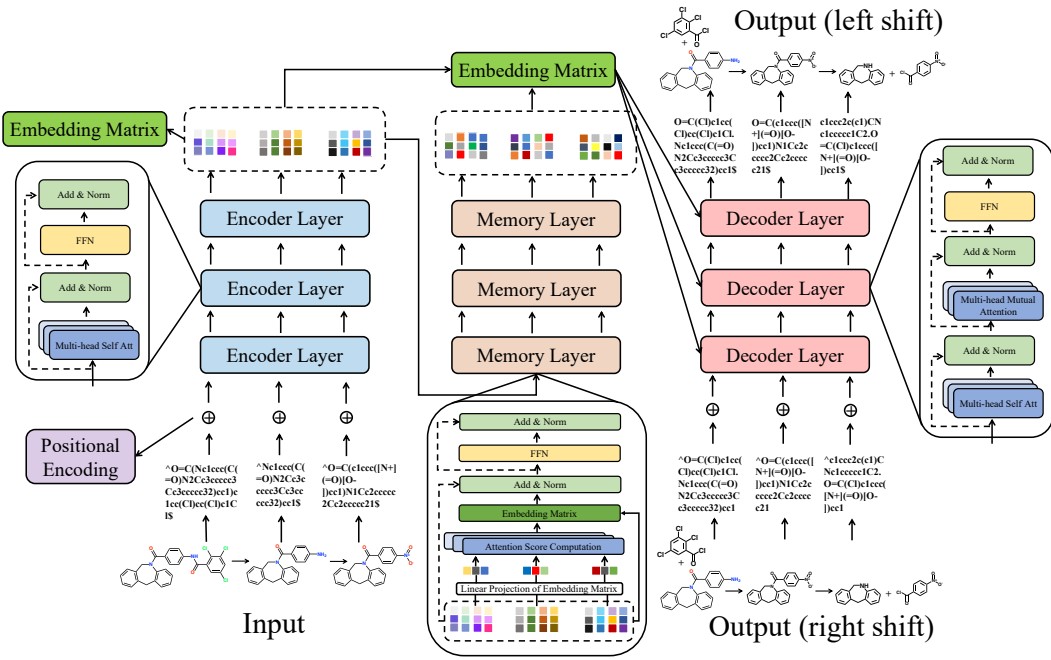

Figure 5: The details of encoder, decoder, and memory module.

# D  DETAILS OF ARCHITECTURE

Fig. 5 outlines the details of three modules: encoder, decoder, and memory module. The encoder and decoder consist of several stacked attention layers. Each attention layer comprises multi-attention heads and a feed-forward layer. The attention heads of each attention layer perform attention mechanism in parallel and then are concatenated and projected into the final embedding. Specifically, an attention head (Scaled Dot-Product Attention) consists of three matrices: the quires $Q$, the keys $K$, and the values $V$. We get the attention scores by multiplying $Q$ and $K$, and then get the output by multiplying the attention scores and $V$. The computation process can be written as

$$\text{Attention}(Q, K, V) = \text{softmax}\left(\frac{QK^T}{\sqrt{d_k}}\right)V, \tag{8}$$

where $\sqrt{d_k}$ is the scaling factor. As illustrated in Fig. 5, we use the self attention layers in the encoder to transform the embeddings of input SMILES into the latent representations, e.g. encoder output. The $Q$, $K$, and $V$ are the same hidden states from the previous layer in the encoder. But for the attention layer in the decoder, $P$ corresponds the embeddings or the hidden states of the output SMILES (right shift), while $Q$ and $V$ correspond the concatenation of the outputs of the encoder and the memory module. RetroXpert Yan et al. (2020) calls this type of attention as encoder-decoder attention. This type of attention enables the decoder to combine the information of the input SMILES and output SMILES to capture the relationship between the product molecule and reactant molecules. By encoding the information of the relationship, we can model the retrosynthesis reaction and make reasonable predictions. Note that the input SMILES given to the decoder are different from the output SMILES predicted by the decoder. The former is the right shift of the output SMILES. In addition, masked attention is used to avoid information leakage during training.

# E  COMPARISON OF DATASETS

We compare our proposed benchmark with Retro* (Chen et al., 2020) and PaRoutes (Genheden & Bjerrum, 2022). As we have discussed in the Introduction, the benchmark of Retro* only consists

of 189 routes in the test set, which is too small to comprehensively evaluate the performance. The test set of our benchmark consists of 5838 reaction trees, which can comprehensively evaluate the performance. PaRoutes does not use the purchasable compounds in eMolecule/ZINC as stop condition. But in our opinion, we think using purchasable compounds in eMolecule/ZINC as stop condition is a better solution with the following reason: If we use the reactions of other patents to build routes, the stock molecules which are constructed with the method proposed in PaRoutes may be different, which is inconsistent and tricky for the evaluation of machine learning models. Using the purchasable compounds in the eMolecule/ZINC databases as the stop condition has better scalability. We can train the machine learning model in one dataset and can apply this model for inference on other datasets directly.

