# OpenReview forum: "Metro: Memory-Enhanced Transformer for Retrosynthetic Planning via Reaction Tree"
_ICLR.cc/2023/Conference — Submitted to ICLR 2023_

### Official Review · Reviewer_tdV3 · 2022-10-22

**Confidence:** 5
**Correctness:** 3
**Technical Novelty And Significance:** 2
**Empirical Novelty And Significance:** 2
**Recommendation:** 3

**Clarity, Quality, Novelty And Reproducibility:**

The paper is clearly written. I think in terms of quality and novelty it is lacking (see details above). Code is not provided, yet the model description is clear and straightforward and one could reimplement it.

**Strength And Weaknesses:**

(+) It is true that existing route data is lacking, so the initiative to propose new data is good. Similarly the idea of including context information makes totally sense to me.

(+) The results look promising.

(-) From an ML perspective, the paper/model contains few novelty.

(-) The paper proposes a new dataset as benchmark and one of the main contributions, but it does neither give much detail about the data nor does it provide a detailed comparison to existing works, apart from two sentences. There are recent datasets and benchmarks, e.g. [1], which should be mentioned and compared to. The related work section does not even contain a paragraph about datasets.

[1] Samuel Genheden and Esben Bjerrum, PaRoutes: towards a framework for benchmarking retrosynthesis route predictions, https://pubs.rsc.org/en/content/articlehtml/2022/dd/d2dd00015f

(-) The evaluation is lacking in that the comparison is with single-step models only. If additional (context) information is included, it is likely that the results will be better. It is interesting to see the magnitude of the improvement, however, to learn about the practical effectiveness I think other experiments are needed. Single-step models are usually applied within a planning algorithm. Since the authors focus on the routes, instead of single steps, I would suggest to evaluate the model also in a more realistic setting. For example, it would be interesting to see how close the proposed model (i.e., simple chaining of the transformer predictions) gets to Retro* with the MLP as single-step model, or how it performs when replacing the MLP by the transformer.

================================================================================

Minor Comments

- sounds wrong to me:
"For two trees ..., we think these are different trees."
"Moreover, what needs to be declared is that molecule synthesis..."
- There are some typos in Algorithm 1
- Why do G2Gs and Graphretro perform that bad?
- What does this mean?:
"2) the dataset is based on reaction route instead of reaction tree."
"Note that in this paper, the default order of the reaction route is the retrosynthesis order."
- The case study in the evaluation, to me it is unclear what it shows specifically, apart from the fact that the proposed model improves upon transformer.

**Summary Of The Paper:**

The paper proposes a new benchmark and transformer model for retrosynthesis prediction. The model particularly makes use of the context of the retrosynthesis routes and is hence rather different from existing ones. It does so by using a memory module and an extension of transformer, using matrices in place of the values (i.e., the value matrix is a tensor here).


**Summary Of The Review:**

The paper proposes a new benchmark and model for retrosynthesis prediction, yet it's missing an evaluation which would justify those.
Altogether, I do not think that the paper can be accepted at the current stage and that it would need quite some revision.

---

> ### Author Response · Authors · 2022-11-05
> **Response to Reviewer tdV3-Part 1**
>
> **Q1:** The concern about the novelty of our proposed model.
> **A1:** We think there is a misunderstanding of the novelty and our contribution. To our best knowledge, we are the first to propose the multi-step retrosynthesis model, which exploits the context information. As we have mentioned in our paper and we also provide the literature [1-30] as follows, existing retrosynthesis models use the product as input for single-step retrosynthesis reaction prediction and do not use context information.
>
>
> **Q2:** The detail about our proposed benchmark.
> **A2:** We use Sec. 4 to describe the details of how to construct our benchmark. Besides, we have provided some background such as reaction route, reaction tree, and starting material to let readers understand our benchmark. We think the reviewer missed these sections.
>
>
> **Single-step Retrosynthesis Model**
>
>
> ***Template-based***
> [1] Retrosim: Computer-assisted Retrosynthesis based on Molecular Similarity, ACS Central Science, 2017.
> [2] Neuralysm: Neural-symbolic Machine Learning for Retrosynthesis and Reaction Prediction, Chemistry–A European Journal, 2017.
> [3] GLN: Retrosynthesis Prediction with Conditional Graph Logic Network, NeurIPS, 2019.
> [4] RetroComposer: Discovering Novel Reactions by Composing Templates for Retrosynthesis Prediction, arXiv, 2021.
>
>
> ***Semi-template-based***
> [5] G2Gs: A Graph to Graphs Framework for Retrosynthesis Prediction, ICML, 2020.
> [6] RetroXpert: Decompose Retrosynthesis Prediction Like A Chemist, NeurIPS, 2020.
> [7] GraphRetro: Learning Graph Models for Retrosynthesis Prediction, NeurIPS, 2021.
> [8] SemiRetro: Semi-template Framework Boosts Deep Retrosynthesis Prediction, arXiv, 2022.
>
>
> ***Template-free***
> [9] Seq2seq: Retrosynthetic Reaction Prediction Using Neural Sequence-to-sequence Models, ACS Central Science, 2017.
> [10] Predicting Retrosynthetic Reactions Using Self-Corrected Transformer Neural Networks, Journal of Chemical Information and Modeling, 2019.
> [11] Transformer: A transformer Model for Retrosynthesis, ICANN, 2019.
> [12] Learning to Make Generalizable and Diverse Predictions for Retrosynthesis, arXiv, 2019.
> [13] Automatic Retrosynthetic Pathway Planning Using Template-free Models, arXiv, 2019.
> [14] RetroPrime: A Chemistry-Inspired and Transformer-based Method for Retrosynthesis Predictions, ChemRxiv, 2020.
> [15] State-of-the-art Augmented NLP Transformer Models for Direct and Single-step Retrosynthesis, Nature Communications, 2020.
> [16] Molecule Edit Graph Attention Network: Modeling Chemical Reactions as Sequences of Graph Edits, ICML GRL+ Workshop, 2020.
> [17] Deep Retrosynthetic Reaction Prediction using Local Reactivity and Global Attention, JACS Au, 2021.
> [18] Dual: Towards Understanding Retrosynthesis by Energy-based Models, NeurIPS, 2021.
> [19] Molecular Graph Enhanced Transformer for Retrosynthesis Prediction, Neurocomputing, 2021.
> [20] Dual-view Molecule Pre-training, arXiv, 2021.
> [21] Valid, Plausible, and Diverse Retrosynthesis Using Tied Two-Way Transformers with Latent Variables, Journal of Chemical Information and Modeling, 2021.
> [22] GTA: Graph Truncated Attention for Retrosynthesis, AAAI, 2021.
> [23] Permutation Invariant Graph-to-sequence Model for Template-free Retrosynthesis and Reaction Prediction, Journal of Chemical Information and Modeling, 2022.
> [24] Retroformer: Pushing the Limits of Interpretable End-to-end Retrosynthesis Transformer, ICML, 2022.
> [25] Chemformer: A Pre-trained Transformer for Computational Chemistry, Machine Learning: Science and Technology, 2022.
> [26] MechRetro is A Chemical-mechanism-driven Graph Learning Framework for Interpretable Retrosynthesis Prediction and Pathway Planning, arXiv, 2022.
> [27] Improving the Performance of Models for One-step Retrosynthesis through Re-ranking, Journal of Cheminformatics, 2022.
> [28] MARS: A Motif-based Autoregressive Model for Retrosynthesis Prediction, arXiv, 2022.
>
>
> ***Selection-based***
> [29] Bayesian Algorithm for Retrosynthesis, Journal of Chemical Information and Modeling, 2020.
> [30] RetCL: A Selection-based Approach for Retrosynthesis via Contrastive Learning, IJCAI, 2021.
>
>
> [31] PaRoutes: towards a framework for benchmarking retrosynthesis route predictions.
> [32] Retro*: Learning Retrosynthetic Planning with Neural Guided A* Search, ICML, 2020.

---

> > ### Author Response · Authors · 2022-11-05
> > **Response to Reviewer tdV3-Part 2**
> >
> > **Q3:** Comparison with other benchmarks.
> > **A3:** We compare our proposed benchmark with Retro* [32] and PaRoutes [31]. As we have discussed in the Introduction, the benchmark of Retro* only consists of 189 routes in the test set, which is too small to comprehensively evaluate the performance. The test set of our benchmark consists of 5838 reaction trees, which can comprehensively evaluate the performance. Note that PaRoutes is a parallel work to ours. We thank the reviewer to point out this paper. We have checked out this paper. PaRoutes does not use the purchasable compounds in eMolecule/ZINC as stop condition. The authors of PaRoutes think "databases such as eMolecules and ZINC are updated with time, making it hard to pick one representative snapshot of the database for the benchmark.". So they instead propose to simply use all the leaves of 10000 routes which they build from USPTO patent as stock molecules. But in our opinion, we think using purchasable compounds in eMolecule/ZINC as stop condition is a better solution with the following reasons:
> >
> > If we use the reactions of other patents to build routes, the stock molecules which are constructed with the method proposed in PaRoutes may be different, which is inconsistent and tricky for the evaluation of machine learning models.
> > Using the purchasable compounds in the eMolecule/ZINC databases as the stop condition has better scalability. We can train the machine learning model in one dataset and can apply this model for inference on other datasets directly.
> > Although the authors of PaRoutes claim that "databases such as eMolecules and ZINC are updated with time, making it hard to pick one representative snapshot of the database for the benchmark.", the update of ZINC has backward compatible treatment as “We also include 40 filtering patterns used in the previous version of ZINC for backward compatibility.” in [38] and "For backwards compatible treatment of ZINC-20 (and ZINC15, ZINC12, etc.) codes, “00” for HAC and LogP indexes indicates this molecule is from one of these older databases." in [39].
> >
> >
> > **Q4:** The concern of the evaluation.
> > **A4:** We think there is a misunderstanding of the evaluation. As we have emphasized in Sec 6, we focus on the evaluation of base models on the retrosynthetic planning task. Besides, some search algorithm papers [32,34,35,36,37] only use one base model and compare the performance.  And the dataset paper [31] also uses one base model for evaluation. Like these papers, we also only use one search algorithm (DFS) and compare the performance of different base models.
> >
> > **Q5:** Some wrong sentences to the reviewer?
> > **A5:** Since there are some different reaction trees that can be used to synthesize the target molecule. We think if one reaction to synthesize the same product between two reaction trees is different, these two reaction trees are different. We show that if the number of steps to synthesize a molecule is larger, the prediction accuracy is lower, which verifies the view that the molecules are hard to synthesize when the number of reaction steps required is larger. So we say "Moreover, what needs to be declared is that molecule synthesis accessibility is a part of our work due to the construction of the dataset."
> >
> > **Q6:** There are some typos in Algorithm 1.
> > **A6:** We think there is a misunderstanding of the inference algorithm. We have provided the code, please refer to the top5_inference.py for more details to understand our inference algorithm.
> >
> > **Q7:** Why do G2Gs and Graphretro perform that bad?
> > **A7:** It seems that the reviewer missed the discussion in Sec. 6.2. As we have emphasized, the results of baselines on our retrosynthetic planning task do not match well with single-step retrosynthesis prediction. Existing semi-template-based models outperform or match the template-based and template-free models on the single-step retrosynthesis prediction but have poor performance on our task. A reasonable reason is that approximately 95% of the reactions in the USPTO-50K dataset used for single-step retrosynthesis reactions have only one reaction center. But in our constructed dataset, approximately 30% of reactions have multiple reaction centers. G2GS only can handle cases with one reaction center, and so perform badly on our task. Template-free models, which do not need to extract templates or find the reaction center using the labeling number, model the retrosynthesis as a sequence-to-sequence problem or a sequence of graph edits, which has better scalability and perform better on retrosynthetic planning tasks.

---

> > > ### Author Response · Authors · 2022-11-05
> > > **Response to Reviewer tdV3-Part 3**
> > >
> > > **Q8:** the dataset is based on reaction route instead of the reaction tree
> > > **A8:** Please refer to our response to your Q3.
> > >
> > > **Q9:** Note that in this paper, the default order of the reaction route is the retrosynthesis order.
> > > **A9:** In general, the order of a reaction route should be from reactants to the product. Since we focus on retrosynthetic planning, so the order of the reaction route is from the product to the reactants.
> > >
> > > **Q10:** The concern about the case study
> > > **A10:** As we have mentioned in the case study, although the reactants that Transformer predicts in the third step are chemically valid, they can not synthesize the product, which means the Transformer makes a wrong prediction on the third-step retrosynthesis reaction and the prediction does not align with the synthesis goal of the current synthetic route. By considering the context information, our proposed model can make correct predictions and discard wrong predictions. In conclusion, the pruning of reaction search space means that we discard some wrong predictions and retain some good predictions for the downstream search algorithms.
> > >
> > > **Q11:** Reproducibility
> > > **A11:** We have provided the code in our general response, if you have any questions about our code, please let us know.
> > >
> > > [34] Self-Improved Retrosynthetic Planning, ICML, 2021.
> > > [35] GNN-Retro: Retrosynthetic Planning with Graph Neural Networks, AAAI, 2022.
> > > [36] RetroGraph: Retrosynthetic Planning with Graph Search, KDD, 2022.
> > > [37] Grasp: Navigating Retrosynthetic Planning with Goal-driven Policy, NeurIPS, 2022.
> > > [38]  ZINC 15 − Ligand Discovery for Everyone.
> > > [39] ZINC-22 - A Free Multi-Billion-Scale Database of Tangible Compounds for Ligand Discovery.

---

> > > > ### Comment · Reviewer_tdV3 · 2022-11-16
> > > > **Thank you for the detailed response.**
> > > >
> > > > **Q1** I see the novelty in that the model uses more data than all the related works you listed, and this is certainly a good idea. My comment referred to the architecture itself, which is a rather standard transformer.
> > > >
> > > > **Q2** Section 4 describes the construction procedure and splits. With a new chemistry dataset, I would expect some information about the nature of the data. As comparison, the OGB graph datasets come with tables about average node degrees, graph diameters etc.
> > > >
> > > > **Q3** Thank you for the comparison, its extent shows in my opinion it's very related work which should not be ignored in the paper.
> > > >
> > > > **Q4**  Maybe I am misunderstanding your goal. How is the model supposed to be used by a chemist?
> > > >
> > > > I do not consider the minor comments critical for a decision, hence I do not comment further on them.

---

> > > > > ### Author Response · Authors · 2022-11-17
> > > > > **Response to your additional questions**
> > > > >
> > > > > **Q1:** I see the novelty in that the model uses more data than all the related works you listed, and this is certainly a good idea. My comment referred to the architecture itself, which is a rather standard transformer.
> > > > > **A1:** Our method uses the standard Transformer to take advantage of context information, and then the performance has been greatly improved, which shows the superiority of our model. We provide a new direction for future multi-step retrosynthesis model design, which is the main contribution of our model. Therefore, we don't limit our model to a specific architecture. We just use the standard Transformer to implement our idea.
> > > > >
> > > > > **Q2:** Dataset statistics
> > > > > **A2:** We provide the graph statistics as follows:
> > > > >
> > > > > | Dataset | # Nodes | # Edges | Average in_degree | Average out_degree |
> > > > > | :--- | :---: | :---: | :---: | :---: |
> > > > > | Reaction Graph | 1125408 | 1665507 | 1.48 | 1.48 |
> > > > >
> > > > > Since the reaction graph is not strongly connected, *networkx* can't return the diameter.
> > > > >
> > > > > **Q3:** The comparison with other datasets.
> > > > > **A3:** Thanks for this suggestion, and we have included the comparison in Appendix in the revised version.
> > > > >
> > > > > **Q4:** How is the model supposed to be used by a chemist?
> > > > > **A4:** We think there is no difference from other single-step retrosynthesis models in terms of usage. In the inference stage, when predicting the reactants, single-step retrosynthesis models only consider the product, and ours need to consider the additional context information (already generated).

---

### Official Review · Reviewer_f3iV · 2022-10-24

**Confidence:** 3
**Correctness:** 1
**Technical Novelty And Significance:** 3
**Empirical Novelty And Significance:** 3
**Recommendation:** 3

**Clarity, Quality, Novelty And Reproducibility:**

## Clarity
As elaborated in the weakness section, there is a room for improvement in terms of clarity.

## Quality
As elaborated in the weakness section, the use of context information is not well motivated to me. At least, it is necessary to compare Metro with existing methods using a variety of search algorithms.

## Novelty
As far as I am aware of, the preprocessing of the USPTO dataset is novel and therefore the proposed setting is novel.

## Reproducibility
Although the hyperparameters are provided in the appendix, the overall learning algorithm is not explicitly described, and it is difficult to identify the exact algorithm. I would suggest the authors to explicitly describe the learning algorithm and/or release the source code to further improve the reproducibility.

**Strength And Weaknesses:**

## Strength
- The experimental results indicate that the proposed method achieves better accuracy scores than baseline methods, including a transformer-based method. The result suggests that the proposed memory module contributes to the performance improvement, validating their approach.
- The authors propose a novel benchmark task considering the limitations of the existing one. In addition, since the details are well described in the paper, it will be not difficult to reproduce the task.

## Weakness
- __Why context information:__ Although the authors discuss why the context information helps in Section 5.3, I am not very convinced of it. The authors claim that the context information can prune the reaction search space, but as far as I understand, many of the existing methods utilize search algorithms for pruning. In fact, all of the methods compared in the experiments use the same DFS algorithm, which is not very fair for the existing methods.
- __Baseline methods do not cover the existing methods introduced in the introduction:__ Since many of the existing methods combine ML and planning algorithms, it is necessary to compare the proposed method with other methods utilizing a wide variety of search algorithms.
- __Presentation needs improvement:__ I had a bit difficulty in understanding the proposed algorithm, because the paper does not provide the whole algorithm (Algorithm 1 was helpful to understand it). In particular, although Figure 2 provides the overview of the proposed method, at first, it is difficult to understand why it has three inputs, A, B, and D. I would appreciate it if the authors could provide the whole algorithm first, and dive into the details after that. Similarly, although the authors pointed out three limitations in the third paragraph of the introduction, they came out suddenly, and I was not very motivated from them.

**Summary Of The Paper:**

This paper is concerned about retrosynthetic planning, where the task is, given a target molecule, to find a reaction tree to synthesize the target molecule from a set of starting compounds. The proposed method called Metro is based on single-step retrosynthesis transformer, but is equipped with a memory module so that it can make use of the retrosynthetic route as context information for the purpose of predicting the reactants. Another contribution is yet another preprocessing of the USPTO dataset, resulting in a new benchmark. The resultant dataset consists of reaction trees extracted from the dataset, and its test dataset is much larger than that in previous benchmark tasks.

The authors conducted empirical studies using the proposed benchmark. In terms of top-$k$ accuracy scores, the proposed method achieves more than 10 points improvements over the transformer model (= Metro - memory module). The authors also confirm that the accuracy degrades as the depth of the reaction tree increases.

**Summary Of The Review:**

I am not positive about accepting this paper, mainly because I am still not motivated to utilize context information. I understand that while it can bias the retrosynthetic process and can prune the search space, its advantage over the existing methods has not been clarified because of the selection of baseline methods. If there is any misunderstanding, I would appreciate it if the authors could point them out.

---

> ### Author Response · Authors · 2022-11-05
> **Response to Reviewer f3iV-Part 1**
>
> **Q1:** Why context information
> **A1:** We think there is a misunderstanding of the pruning of reaction search space and our contribution of using context information. As discussed in Sec. 2.1 in [1], the search algorithm* uses the reaction cost to quality the solution. But they still rely on the base model ( template-based MLP model in their paper) to make single-step retrosynthesis reaction predictions. As we all know, different base models have different performances on the single-step retrosynthesis prediction task.  As we have mentioned in the case study, although the reactants that Transformer predicts in the third step are chemically valid, they can not synthesize the product, which means the Transformer makes a wrong prediction on the third-step retrosynthesis reaction and the prediction does not align with the synthesis goal of the current synthetic route. By considering the context information, our proposed model can make correct predictions and discard wrong predictions. In conclusion, the pruning of reaction search space means that we discard some wrong predictions and retain some good predictions for the downstream search algorithms.
>
> **Q2:** In fact, all of the methods compared in the experiments use the same DFS algorithm, which is not very fair for the existing methods.
> **A2:** We think there is a misunderstanding of fair comparison. As shown in the search algorithm papers [1,2,3], while comparing the performance of different search algorithms, they also used only one base model. As we have mentioned in this paper, we focus on the evaluation of base models on the retrosynthetic planning task. We only use one memory module on top of the transformer to exploit context information and do not use any search algorithm components in our training and model. All the base models are trained using the same training dataset and evaluated using the same search algorithm.
>
> **Q3:** Baseline methods do not cover the existing methods introduced in the introduction
> **A3:** As we have emphasized in the experimental section, we focus on the evaluation of the base models on the retrosynthetic planning task. Our model outperforms baselines by at least 10.7% of top-1 test accuracy. Such a large margin is able to fully demonstrate that our model is much better than baselines. As the reviewer is interested, we will try our best to conduct experiments with other search algorithms, such as BFS during the rebuttal phase. But we can imagine the workload is so huge since we need to conduct experiments with 7 baselines. Besides, we also need to write the codes for the search algorithm and many of them [2, 3] do not release the codes. And we also need to train the search algorithm.
>
> **Q4:** The understanding of our algorithm
> **A4:** We also provide the details of the architecture in Append D in the revised version of our paper. As we have mentioned in Sec. 5.1.2, we model the the prediction of reaction route as a sequence of molecule generation. So the input is the reaction route. Given the SMILES representations of (A, B, D) from the same reaction route as input, the output of our model is the SMILES representations of (B+C, D, E+F). When we predict B+C, the input is A and we will not consider the information of (B, D). When we complete the prediction of (B, C), we add B to the reaction route and use (A, B) as input to predict D. We repeat this step until all reactants are starting materials. So during the training stage, we do not consider the information of later molecules when making the current prediction. The attention mechanism ensures that our predictions of all reactions along the same route are parallelized at the training stage. This can be implemented by masking the input of later molecules. You also can find these details in our released codes. Algorithm 1 outlines the process of the inference stage.
>
>
> [1] Retro*: Learning Retrosynthetic Planning with Neural Guided A* Search
> [2] RetroGraph: Retrosynthetic Planning with Graph Search
> [3] GRASP: Navigating Retrosynthetic Planning with Goal-driven Policy
> [4] G2Gs: A Graph to Graphs Framework for Retrosynthesis Prediction

---

> > ### Author Response · Authors · 2022-11-05
> > **Response to Reviewer f3iV-Part 2**
> >
> > **Q5:** About three limitations.
> > **A5:** We are sorry for this.
> > 1) As we have discussed in the Abstract, the multi-step reactions are crucial because they determine the flow chart in the production of the Organic Chemical Industry. However, existing datasets lack curation of tree-structured multi-step reactions and fail to provide such reaction trees, limiting models’ understanding of organic molecule transformations.
> > 2) Past works neglect the context information of the reaction tree. Since they use the single-step retrosynthesis models as base models. To our best knowledge, we are the first to propose the multi-step retrosynthesis model to exploit the context information. [4] also points out one future work is to extend G2Gs to cope with multi-step retrosynthesis tasks. So we need consider the context information for the design of multi-step retrosynthesis models.
> > 3) Retro* only uses 189 test routes for evaluation. The test set is too small to comprehensively evaluate the performance.

---

### Official Review · Reviewer_1qQ1 · 2022-10-24

**Confidence:** 4
**Correctness:** 3
**Technical Novelty And Significance:** 3
**Empirical Novelty And Significance:** 3
**Recommendation:** 5

**Clarity, Quality, Novelty And Reproducibility:**

## Clarity
The paper is written quite well, but needs further details in aforementioned topices

## Quality
OK

## Novelty
The dataset is novel, as well as the application of a Memory Module of reaction-trees

## Reproducibility
No code provided

**Strength And Weaknesses:**

## Strengths:
 - Propose a new Benchmark for Multi-Step-Retrosynthesis
 - Propose Metro: Memory-Enhanced Transformer for RetrOsynthetic planning by extending Transformer with an additional memory module
    - reaction route as context information
    - can control the synthesis path

## Weaknesses:
 - Missing Important Related Work:
   - Section for Memory Enhanced Architectures, completely missing [1]
   - Missing Template-Based Methods in Related Works section [2] [3]
 - More clarification on why extracted Reaction-Trees are seen as optimal
 - No code for reproducibility is provided
 - Architecture needs further explanations, e.g. Decoder, Tokenizer, "Concatenated Embedding" + residual connection, ...
 - The proposed dataset is a subset of USPTO-Full, please provide a comparison e.g. a scatter plot, of performance differences between e.g. USPTO-Full results and the results obtained on your benchmark set

## Minor Comments
 - Was SMILES augmentation [4] performed or any other type of augmentation?
 - 5.1.1 paragraph about SMILES representation 2 might fit more in the subsequent paragraph 5.1.2, but is nothing inherent about the architecture
 - Please provide variance for point estimate in Table 1 as well as the significance
 - Is any standardization performed on molecules?
 - Please further describe the meaning behind A B C D E F in Figure 2
 - Appendix B.2.: Also describe the hyperparameter search space, as well as how they were obtained
 - Further discussion on why extracted Reaction-Trees are seen as optimal/better; e.g. there might exist faster/more optimal reaction routes

### References
 - [1] Ramsauer, H., Schäfl, B., Lehner, J., Seidl, P., Widrich, M., Adler, T., ... & Hochreiter, S. (2020). Hopfield networks is all you need.
 - [2] Chen, S., & Jung, Y. (2021). Deep retrosynthetic reaction prediction using local reactivity and global attention. JACS Au, 1(10), 1612-1620.
 - [3] Seidl, P., Renz, P., Dyubankova, N., Neves, P., Verhoeven, J., Wegner, J. K., ... & Klambauer, G. (2021). Modern Hopfield Networks for Few-and Zero-Shot Reaction Prediction.
 - [4] Tetko, I. V., Karpov, P., Van Deursen, R., & Godin, G. (2020). State-of-the-art augmented NLP transformer models for direct and single-step retrosynthesis. Nature communications, 11(1), 1-11.

**Summary Of The Paper:**

The authors propose a transformer with memory of reaction-trees for a new multi-step-retrosynthesis benchmark.

**Summary Of The Review:**

The paper introduces a new interesting benchmark for multi-step retrosynthesis that might be influential in the community of retrosynthesis, however no code or dataset is provided.
The architecture seems interesting, but lacks connection to related work, as well as further clarifications.

---

> ### Author Response · Authors · 2022-11-05
> **Response to Reviewer 1qQ1-Part 1**
>
> **Q1:** Missing Important Related Work
> **A1:** Thanks for pointing out the related work and we will cite this paper.  Please see the revised version of our paper.
>
>
> **Q2:** More clarification on why extracted Reaction-Trees are seen as optimal
> **A2:** [1] construct a knowledge graph and uses the depth as a metric to quantify the synthesizability. In their opinion, the drug molecules that require longer routes to synthesize are hard-to-synthesize in general.  Besides, PaRoutes [2] also utilizes depth as a metric for the evaluation of reaction trees. 1) "We then discarded all routes with a depth of more than 10 reactions, to exclude a few really long routes that will require an extensive search (this excludes less than 100 out of the 150 K routes, possibly reflecting USPTOs patent oriented chemical scope without extensive coverage of complex syntheses). " 2) "The dataset is tilted towards short routes with few leaves."  So, by extracting reaction trees with minimum depth, we can get the optimal reaction trees.
>
>
> **Q3:** No code for reproducibility is provided
> **A3:** We have provided the code link in the general response.
>
> **Q4:** Architecture needs further explanations, e.g. Decoder, Tokenizer, "Concatenated Embedding" + residual connection, ...
> **A4:** We are sorry for this. As we have mentioned, the architectures of the encoder and decoder are the same as [3]. In order to allow reviewers to understand our architecture, we provide the architecture details in Appendix D.
>
> **Q5:** The proposed dataset is a subset of USPTO-Full, please provide a comparison e.g. a scatter plot, of performance differences between e.g. USPTO-Full results and the results obtained on your benchmark set.
> **A5:** We extract the USPTO-Full results from the GLN [4] paper.
> USPTO-Full
> |  | retrosim | neuralsym | GLN |
> | :---: | :---: | :---: | :---: |
> | top-1 | 32.8 | 35.8 | 39.3 |
> | top-10 | 56.1 | 60.8 | 63.7 |
>
>
> The results obtained on our benchmark set.
> |  | retrosim | neuralsym | GLN |
> | :---: | :---: | :---: | :---: |
> | top-1 | 23.2 | 26.8 | 25.9 |
> | top-5 | 30.5 | 35.8 | 37.2 |
>
> Note that the first is the results of a single-step retrosynthesis reaction prediction, and the second is the results of our retrosynthesis planning task.
>
>
> [1] Prediction of compound synthesis accessibility based on reaction knowledge graph.
> [2] PaRoutes: towards a framework for benchmarking retrosynthesis route predictions.
> [3] A transformer model for retrosynthesis.
> [4] Retrosynthesis Prediction with Conditional Graph Logic Network.

---

> > ### Author Response · Authors · 2022-11-05
> > **Response to Reviewer 1qQ1-Part 2**
> >
> > **Q6:** Was SMILES augmentation [4] performed or any other type of augmentation?
> > **A6:** No, we do not use any data augmentation.
> >
> > **Q7:** 5.1.1 paragraph about SMILES representation 2 might fit more in the subsequent paragraph 5.1.2, but is nothing inherent about the architecture
> > **A7:** We think there is a misunderstanding of the purpose of this paragraph. In Sec. 5.1.1, we introduce the input, output, and model from a high-level perspective. The input for our model is the SMILES representations of molecules. And there is a difference between the output SMILES fed into the decoder and the output SMILES predicted by the decoder. We add the start symbol $\boldsymbol{\wedge}$ (right shift) on the former output SMILES and the end symbol $ (left shift) on the later output SMILES. So there is a need to use this paragraph to introduce these details. In Sec. 5.1.2, we use this paragraph to model our task. We model the prediction of the reaction route as a sequence of molecule generation.
> >
> > **Q8:** Is any standardization performed on molecules?
> > **A8:** No
> >
> > **Q9:** Please further describe the meaning behind A B C D E F in Figure 2
> > **A9:** The A B C D E F corresponds to the molecule in the reaction tree as shown in Figure. 1. As discussed in the caption of Figure 1, A is the desired product molecule to be synthesized. B, C, and D are the intermediate product molecules. E, F, G, and H are the starting molecules.
> >
> > **Q10:** Appendix B.2.: Also describe the hyperparameter search space, as well as how they were obtained
> > **A10:** As we have discussed in the Implementation Details Paragraph, we only introduce additional 3 memory layers on top of the Transformer and do not touch other components. For the hyper-parameters, we directly followed the reported setting in their released codes and did not perform additional hyperparameter tuning. We still achieve a dramatic improvement in the planning task, which well demonstrates the effectiveness of our proposed method.
> >
> > **Q11:**  Further discussion on why extracted Reaction-Trees are seen as optimal/better; e.g. there might exist faster/more optimal reaction routes
> > **A11:** Please the response A2 to your Q2.

---

### Official Review · Reviewer_r9kB · 2022-11-05

**Confidence:** 5
**Correctness:** 3
**Technical Novelty And Significance:** 2
**Empirical Novelty And Significance:** 1
**Recommendation:** 3

**Clarity, Quality, Novelty And Reproducibility:**

### quality
- the paper has several incorrect definitions.
  - in chemistry, a synthetic route and synthetic tree are the same
  - the depth of a tree is not particularly important, what is more important is the total number of steps in the tree. with this definition, the authors are solving a different task previous work, and comparison to previous work is therefore not very meaningful, as a different metric was used
  - it is unclear why a transformer model is needed. RETROGRAPH https://arxiv.org/abs/2206.11477 shows a transformer is not needed for adding in contextual information in retrosynthesis search
- the molecular case study contains incorrect reactions, which casts doubt on the usefulness of the method in practice
- the results in table 1 are inconsistent with previous works (e.g. GLN should be stronger than neuralsym, single step transformer stronger than neuralsym and GLN). it is unclear whether the baselines have been tuned properly.
### clarity
good

### originality
1) The benchmark proposed in this paper is very close to the PAroutes  benchmark, which was already published in Feb 2022 https://chemrxiv.org/engage/chemrxiv/article-details/621c86f3c3e9da4f737b0047 - why do we need yet another benchmark?
2) The use of graph and contextual information has already been proposed in RETROGRAPH https://arxiv.org/abs/2206.11477 earlier in 2022

**Strength And Weaknesses:**

### strengths
- clear description
- interesting memory model

### weaknesses:
- the paper has several incorrect definitions
- the paper yet proposes another benchmark for retrosynthesis even though there are already several existing ones
- the presented model is not properly compared
- the baselines in the paper are inconsistent with previous work
- somewhat limited ML model novelty
- some proposed reactions in the case study are chemically incorrect

**Summary Of The Paper:**

- a benchmark for retrosynthesis is proposed
- a model for retrosynthesis is presented
- the authors claim state of the art

there are only limited novel contributions in this paper

**Summary Of The Review:**

The proposed benchmark and approach are a small variation of previously existing work. Therefore this reviewer cannot justify to recommend  this work for publication in a top ML conference.

---

> ### Author Response · Authors · 2022-11-05
> **Response to Reviewer r9kB-Part1**
>
> **Q1:** About the benchmark
> **A1:** We compare our proposed benchmark with Retro* [1] and PaRoutes [2]. As we have discussed in the Introduction, the benchmark of Retro* only consists of 189 routes in the test set, which is too small to comprehensively evaluate the performance. The test set of our benchmark consists of 5838 reaction trees, which can comprehensively evaluate the performance. Note that PaRoutes is a parallel work to ours. We thank the reviewer to point out this paper. We have checked out this paper. PaRoutes does not use the purchasable compounds in eMolecule/ZINC as stop condition. The authors of PaRoutes think "databases such as eMolecules and ZINC are updated with time, making it hard to pick one representative snapshot of the database for the benchmark.". So they instead propose to simply use all the leaves of 10000 routes which they build from USPTO patent as stock molecules. But in our opinion, we think using purchasable compounds in eMolecule/ZINC as stop condition is a better solution with the following reasons:
> 1) If we use the reactions of other patents to build routes, the stock molecules which are constructed with the method proposed in PaRoutes may be different, which is inconsistent and tricky for the evaluation of machine learning models.
> 2) Using the purchasable compounds in the eMolecule/ZINC databases as the stop condition has better scalability. We can train the machine learning model in one dataset and can apply this model for inference on other datasets directly.
> 3) Although the authors of PaRoutes claim that "databases such as eMolecules and ZINC are updated with time, making it hard to pick one representative snapshot of the database for the benchmark.", the update of ZINC has backward compatible treatment as “We also include 40 filtering patterns used in the previous version of ZINC for backward compatibility.” in [3] and "For backwards compatible treatment of ZINC-20 (and ZINC15, ZINC12, etc.) codes, “00” for HAC and LogP indexes indicates this molecule is from one of these older databases." in [4].
>
> **Q2:** About the evaluation
> **A2:** We think there is a misunderstanding of our evaluation.  As we have emphasized in Sec 6, we focus on the evaluation of base models on the retrosynthetic planning task. Besides, some search algorithm papers only use one base model and compare the performance. And the PaRoutes [1] also exploits one base model for evaluation. Like these papers, we also only use one search algorithm (DFS) and compare the performance of different base models. All the base models are trained and tested with the same dataset. And all the base models exploit the search algorithm for search. We do not see any unfair comparison.
>
> **Q3:** About the novelty of our model
> **Q3:** We think there is a misunderstanding of the novelty and our contribution. To our best knowledge, we are the first to propose the multi-step retrosynthesis model, which exploits the context information. As we have mentioned in our paper and we also provide the literature in our response to Reviewer tdV3, existing retrosynthesis models use the product as input for single-step retrosynthesis reaction prediction and do not use context information.
>
> **Q4:** About the case study
> **A4:** The case study is constructed by the predictions of our model and Transformer. The chemically incorrect reactions are from the predictions of the Transformer, which indicates the Transformer makes a wrong prediction on the third-step retrosynthesis reaction and the prediction does not align with the synthesis goal of the current synthetic route. By considering the context information, our proposed model can make correct predictions and discard wrong predictions. In conclusion, the pruning of reaction search space means that we discard some wrong predictions and retain some good predictions for the downstream search algorithms.
>
> **Q5:** in chemistry, a synthetic route and synthetic tree are the same
> **A5:** Maybe a synthetic route and a synthetic tree are the same in chemistry.  But in Computer Science, routes are just parts of a tree. ICLR is a machine learning conference. The background of the readers is also usually CS/EE. We need to make clearer definitions so that readers don't get confused.

---

> > ### Author Response · Authors · 2022-11-05
> > **Response to Reviewer r9kB-Part2**
> >
> > **Q6:** the depth of a tree is not particularly important
> > **A6:** In fact, PaRoutes also utilizes depth as a metric for the evaluation of reaction trees. 1) "We then discarded all routes with a depth of more than 10 reactions, to exclude a few really long routes that will require an extensive search (this excludes less than 100 out of the 150 K routes, possibly reflecting USPTOs patent oriented chemical scope without extensive coverage of complex syntheses). " 2) "The dataset is tilted towards short routes with few leaves." [5] construct a knowledge graph and uses the depth as a metric to quantity the synthesizability. In their opinion, the drug molecules that require longer routes to synthesize are hard-to-synthesize in general.
> >
> > **Q7:** The difference between our model and RetroGraph
> > **A7:** RetroGraph is a search algorithm, and ours is a base model.  By exploiting the context information in the base model, we can make some good predictions for the downstream search algorithms on the retrosynthetic planning task. Moreover, to compute the regression loss, Retro* uses the shortest routes to compute the ground truth cost for the target molecules, which also exploits the context information. So RetroGraph is not the first search algorithm to use context information.
> >
> > **Q8:** the results in table 1 are inconsistent with previous works
> > **A8:** We use the parameters provided in their released codes, please refer to our released codes in the general response for more details.
> >
> > [1] Retro*: Learning Retrosynthetic Planning with Neural Guided A* Search.
> > [2] PaRoutes: towards a framework for benchmarking retrosynthesis route predictions.
> > [3] ZINC 15 − Ligand Discovery for Everyone.
> > [4] ZINC-22 - A Free Multi-Billion-Scale Database of Tangible Compounds for Ligand Discovery.
> > [5] Prediction of compound synthesis accessibility based on reaction knowledge graph.

---

> ### Author Response · Authors · 2022-12-01
> **We need your feedback on our response.**
>
> Dear Reviewer r9kB,
> We need your feedback on our response.  And regarding the metric to evaluate the reaction tree, we find Retro* also uses depth. You can find the details in their paper in Section 5.1.
>
> *Given the list of building blocks, we take each molecule that have appeared in USPTO reaction data and analyze if it can be synthesized by existing reactions within USPTO training data. For each synthesizable molecule, we choose the shortest-possible synthesis routes with ending points being available building blocks in eMolecules.* [1]
>
> In fact, the depth of a reaction tree is highly correlated with the number of reaction steps. With shorter reaction trees, we can synthesize the target molecules in a shorter time. Because reactions at the same depth can be processed parallelized.
>
> [1] Retro*: Learning Retrosynthetic Planning with Neural Guided A* Search.
> Best,
> Authors

---

### Author Response · Authors · 2022-11-05
**General response to the reviews**

First of all, we thank the reviewers for their valuable comments and feedback. In summary, reviewers generally agree that our proposed benchmark is novel and new. They appreciate that "The authors propose a novel benchmark task considering the limitations of the existing one" and "It is true that existing route data is lacking, so the initiative to propose new data is good". Besides, they think "the experimental results indicate that the proposed method achieves better accuracy scores than baseline methods". In the following, we address the concerns of reviewers.

We provide the code and the datasets at this link: https://anonymous.4open.science/r/metro-BAB5/.

---

### Decision · Program_Chairs · 2023-01-20

**Decision:**

Reject

**Justification For Why Not Higher Score:**


Although the paper indeed contains new insight for retrosynthesis with new benchmarks, the current version of the paper does not demonstrate the significance of the contributions among the existing literature, which makes the paper less interesting.

I believe more effort is needed to improve the paper.

**Justification For Why Not Lower Score:**

N/A

**Metareview: Summary, Strengths And Weaknesses:**


In this paper, the authors constructed a retrosynthesis benchmark, which includes the multi-step retrosynthesis path as the prediction target. The authors also proposed a memory-enhanced transformer model, which empirically achieve strong empirical performances.

The major concerns raised by most of the reviewers lies in the following aspects:

- The novelty of the model. The model architecture is relatively straightforward.

- The presentation of the paper is not clear, making the paper difficult to follow.

- The literature is not fully covered, including the methods proposed in chemistry, which makes the paper is not well-positioned.

- The experiments results are not well presented, which cause confusion and insufficient comparison impression from reviewers.


I do believe the paper contains contribution to the community. However, the current version does not clearly emphasize the novelty and significance. I suggested the authors can consider the comments provided by the reviewers to improve the draft.